# Impact of Low-Head Dam Removal on River Morphology and Habitat Suitability in Mountainous Rivers

**DOI:** 10.3390/ijerph191811743

**Published:** 2022-09-17

**Authors:** Yun Lu, Wan-Yi Zhu, Qing-Yuan Liu, Yong Li, Hui-Wu Tian, Bi-Xin Cheng, Ze-Yu Zhang, Zi-Han Wu, Jie Qing, Gan Sun, Xin Yan

**Affiliations:** 1State Key Laboratory of Hydraulics and Mountain River Engineering, Sichuan University, Chengdu 610065, China; 2Chengdu Xingcheng Capital Management Co., Ltd., Chengdu 610000, China; 3Yangtze River Fisheries Research Institute of Chinese Academy of Fisheries Science, Wuhan 430223, China; 4Shanghai Investigation, Design and Research Institute Corporation Limited, Shanghai 200434, China; 5China Three Gorges Construction (Group) Co., Ltd., Chengdu 610041, China

**Keywords:** dam removal, fish reproduction, habitat suitability, morphology, assessment

## Abstract

Dam removal is considered an effective measure to solve the adverse ecological effects caused by dam construction and has started to be considered in China. The sediment migration and habitat restoration of river ecosystems after dam removal have been extensively studied abroad but are still in the exploratory stage in China. However, there are few studies on the ecological response of fishes at different growth stages. Considering the different habitat preferences of *Schizothorax prenanti* (*S. prenanti*) in the spawning and juvenile periods, this study coupled field survey data and a two-dimensional hydrodynamic model to explore the changes in river morphology at different scales and the impact of changes in hydrodynamic conditions on fish habitat suitability in the short term. The results show that after the dam is removed, in the upstream of the dam, the riverbed is eroded and cut down and the riverbed material coarsens. With the increase in flow velocity and the decrease in flow area, the weighted usable area (WUA) in the spawning and juvenile periods decreases by 5.52% and 16.36%, respectively. In the downstream of the dam, the riverbed is markedly silted and the bottom material becomes fine. With the increase in water depth and flow velocity, the WUA increases by 79.91% in the spawning period and decreases by 67.90% in the juvenile period, which is conducive to adult fish spawning but not to juvenile fish growth. The changes in physical habitat structure over a short time period caused by dam removal have different effects on different fish development periods, which are not all positive. The restoration of stream continuity increases adult fish spawning potential while limiting juvenile growth. Thus, although fish can spawn successfully, self-recruitment of fish stocks can still be affected if juvenile fish do not grow successfully. This study provides a research basis for habitat assessment after dam removal and a new perspective for the subsequent adaptive management strategy of the project.

## 1. Introduction

The excessive and disorderly development of small hydropower groups is the main reason for the habitat degradation and fragmentation of tributary river networks. Dams hinder the migration route of fish, change the runoff process and lead to the degradation and loss of river habitats. According to the IUCN (International Union for Conservation of Nature), dams have been the main reason for the extinction or endangerment of nearly one-fifth of the fish species worldwide in the past 100 years [1,2]. With a large number of small dams reaching the end of their design life and their management and maintenance requirements not being met, potential safety hazards are becoming increasingly prominent. Under the condition that maintenance and reinforcement still cannot solve their potential safety hazards, the removal of low-head dams as an eco-friendly strategy has attracted increasing attention [3,4]. The effect of dam removal on restoring the interaction between geomorphic and hydrological processes and on reshaping the river channel through erosion and sedimentation processes have been well discussed [5,6,7,8,9]. In addition, dam removal has been proven to be an effective way to increase the diversity of river ecosystems, as it reconnects rivers and promotes gene exchange between populations [10,11,12]. For example, O’Donnell et al. [13] found that within one year after the removal of the Edwards dam on the Kennebec River, a large number of American eel (*Anguilla rostrata*), Alewife (*Alosa pseudoharengus*) and Atlantic sturgeon (*Acipenser oxyrinchus*) were observed in the upstream habitat, which had disappeared for over 150 years. Catalano et al. [14] and Kanehl et al. [15] found that after dam removal, the static water environment becomes a torrent environment, and the number of shoals, bends and torrents increases. Fish that prefer river habitats gradually replace fish that prefer lake habitats. However, although studies on the habitat and ecological response after dam removal have been carried out in recent years, as mentioned above, most of them have only focused on the impact of dam removal on adult fish, especially on spawning activities [6,16,17,18]. Indeed, habitat preference is far more complicated for fish at different periods, and the impact of physical habitat changes on other sensitive growth periods is less studied.

The replenishment of fish populations is highly dependent on habitat availability and habitat quality [19,20,21,22]. Only when external environmental conditions such as water temperature, hydrology and hydraulic power are met and sufficient energy is accumulated can fish complete the corresponding life activities [23,24]. Spawning activity, which bonds parents and offspring and is believed to be key to boosting stocks, has been well discussed by scholars [25,26,27,28]. Shen et al. [29] found that the delay of the spawning time of Chinese sturgeons was attributed to the increase in water temperature and the decrease in downstream discharge through model simulation. Warren et al. [30] reviewed the impact of flow on the number and distribution of salmon and found that improper hydraulic and ecological hydrological conditions delay the spawning time and reduce the number of eggs, which may further lead to the aging of fish populations and reduce the replenishment of fish resources. The juvenile period, where fish begin to store energy by increasing feeding for the maturation of internal organs, is another critical period. Although the living habits of juvenile fish are roughly the same as those of adult fish, studies have shown that environmental interference may lead to the death of a large number of juvenile fish, meaning that they are more sensitive to environmental changes than adult fish [31,32,33]. As a major source of parental supplementation, the successful growth of juvenile fish is very important to maintain the sustainable regeneration of fishery resources so fish can mate once sexually mature [34,35]. However, the habitat needs of juvenile fish are usually ignored in the study of river ecological restoration. With the development of environmental awareness in various countries, higher requirements have been put forward for the protection of fish diversity, which highlights the need for a deeper understanding of the potential role of the removal of low-head dams in the protection and restoration of key habitats.

At present, the impact assessment of dam removal is mainly carried out via two different approaches: actual monitoring data analysis and theoretical model prediction [36]. The site monitoring and assessment of dam removal cases is the most direct and convincing research method, but due to cost and some noncontrollable factors, monitoring data are sometimes not comprehensive. Numerical simulation can predict the ecological impact of dam removal because of its fast speed, low cost and strong simulation ability. However, sometimes the accuracy of the simulation cannot be guaranteed [37]. In recent years, both methods have been widely used in the ecological impact assessment of dam removal [38,39]. Comprehensively considering the advantages and disadvantages of the two evaluation methods, the best way to evaluate the ecological impact and provide a reliable basis for the dam removal decision is to use the numerical simulation method to quickly grasp the changes in fish habitat in different spatial locations, determine the impact of dam removal on fish and use the monitoring data to provide evidence for the simulation results and to verify the correctness of the model-based analysis.

Taking the Laomuhe Dam removal project of the Heishui River as an example, this study investigates the impact of dam removal on river morphology and fish critical habitats in the short term (within three years after dam removal) by incorporating the juvenile and spawning habitat requirements into the ecological impact assessment of dam removal by coupling the numerical simulation and field survey methods. It further discusses how to develop and manage river ecological restoration measures to more comprehensively maintain river fish habitat quality at a high level in small and medium mountainous rivers. Compared with our previous research results in the Heishui River [23,40], this study is a continuation and extension of the research on the impact of external environmental changes on fish. The research results can provide theoretical support for the environmental impact assessment and adaptive management of subsequent river restoration projects.

## 2. Materials and Methods

### 2.1. Study Area

The Heishui River is the first-class tributary on the left bank of the Jinsha River and a typical mountain river in Southwest China. With a total length of 173.0 km, the riverbed bottom is mainly composed of pebbles and gravel, with occasional large boulders. There are four cascade dams in the Heishui River, namely Sujiawan Dam (SJWD), Gongdefang Dam (GDFD), Songxin Dam (SXD) and Laomuhe Dam (LMHD), from upstream to downstream. The four dams are all run-of-river hydropower stations without regulation capability (Figure 1).

The LMHD (102°38’ E, 27°10’ N) is 7.8 m high and 56 m wide. Water diversion has greatly changed the natural hydrological conditions of the river section, making the downstream river section severely dehydrated. As a result, the biological habitat in this area has been severely degraded, and there are great differences in fish population structures between the upstream and downstream sections. To repair the local harsh habitats, restore river connectivity and extend the length of natural river sections, the LMHD was removed in December 2018. The study area of this study is the river section where the habitat quality is severely disturbed between 1 km upstream and 0.5 km downstream of the LMHD. According to the flow monitoring data of Ningnan hydrological station for the last 20 years, under natural conditions, the average flow during the centralized spawning period from March to May was 23.55 m^3^/s. After the completion of the LMHD, the average flow downstream of the dam was 2.95 m^3^/s (Figure 1).

### 2.2. Field Data Collection and Target Fish

Since December 2018, 25 monitoring sections along the main stream of the Heishui River have been tracked in spring and autumn every year (organized by Shanghai Investigation, Design and Research Institute Corporation Limited, Shanghai, China). Four monitoring sections are included in the study area of this manuscript. The four monitoring sections are located 1 km upstream of LMHD, 400 m upstream of LMHD, LMHD site and 400 m below the LMHD. The monitoring content includes velocity, water depth, discharge, riverbed substrate and channel topography. Hydrodynamic indexes (velocity and water depth) were performed by Acoustic Doppler Current Profiler (ADCP), portable flow rate meter and depth sounder RTK according to the actual situation of the river. During each tracking and monitoring, the sand shovel was used to collect appropriate riverbed materials at each monitoring section and put them into plastic buckets. After drying in the laboratory, electric mesh screen was used to screen the bed material with different particle sizes, and the weight of sediment with different particle sizes was recorded. Finally, the percentage was calculated and the sediment grain grading curve was drawn. In addition, a total of six automatic monitoring stations were set up in the Heishui River (Figure 1), and each station monitored the flow in the reach on a daily basis to provide the required flow data for the numerical simulation of the study. In order to understand the changes in riverbed topography before and after the removal of LMHD, the riverbed and the topography of both sides of the study area were mapped in May 2018, May 2019 and May 2021, respectively. The survey contents included river topography and sectional topography. According to the measurement data of river elevation in the study area, the measured data of limited river section topography were used to construct refined river topography by encrypting sections and interpolation, which was used to analyze the changes in river topography before and after dam removal.

According to the preliminary survey results of fish resources, Schizothorax prenanti (*S. prenanti*) is the main dominant fish in the Heishui River [41,42]. Due to overfishing, hydropower development and water pollution, its resources have decreased significantly in recent years [43]. The Sichuan provincial government has listed *S. prenanti* as a priority protected species. Considering the economic status and degree of endangerment of *S. prenanti* in the Heishui River, this study selected *S. prenanti* as the target fish [40,44]. *S. prenanti* is a typical plateau fish species that mainly lives in mountainous rivers with fast flow velocity and high oxygen content and exhibits short-distance reproductive migration. The breeding period of *S. prenanti* in the study area is from December to May, during which time the peak spawning period is from March to May. Its eggs are heavy and sticky and are mostly deposited on the gravel of the rapids [42].

### 2.3. Habitat Simulation Model

#### 2.3.1. Hydrodynamic Model

The two-dimensional hydrodynamic model of MIKE is applied to explore the changes in hydrodynamic conditions before and after dam removal. The hydrodynamic model of the research is based on the Navier–Stokes equation with three-dimensional incompressibility and uniform Reynolds values, and is subject to the Boussinesq assumption and the assumption of hydrostatic pressure [45,46,47]. The shallow water equations, including continuity and momentum equations, are given by
(1)$$\frac{\partial h}{\partial t}+\frac{\partial h\bar{u}}{\partial x}+\frac{\partial h\bar{v}}{\partial y}=hS$$
(2)$$\begin{array}{l} \frac{\partial h\bar{u}}{\partial t}+\frac{\partial h{\bar{u}}^{2}}{\partial x}+\frac{\partial h\bar{vu}}{\partial y}=f\bar{v}h-gh\frac{\partial \eta }{\partial x}-\frac{h}{\rho_{0}}\frac{\partial p_{a}}{\partial x}-\frac{gh^{2}}{2\rho_{0}}\frac{\partial \rho }{\partial x}+\frac{\tau_{sx}}{\rho_{0}}-\frac{\tau_{bx}}{\rho_{0}}\\ -\frac{1}{\rho_{0}}\left(\frac{\partial s_{xx}}{\partial x}+\frac{\partial s_{xy}}{\partial y}\right)+\frac{\partial }{\partial x}\left(hT_{xx}\right)+\frac{\partial }{\partial y}\left(hT_{xy}\right)+hu_{s}S \end{array}$$
(3)$$\begin{array}{l} \frac{\partial h\bar{v}}{\partial t}+\frac{\partial h\bar{uv}}{\partial x}+\frac{\partial h{\bar{v}}^{2}}{\partial y}=-f\bar{u}h-gh\frac{\partial \eta }{\partial y}-\frac{h}{\rho_{0}}\frac{\partial p_{a}}{\partial y}-\frac{gh^{2}}{2\rho_{0}}\frac{\partial \rho }{\partial y}+\frac{\tau_{sy}}{\rho_{0}}-\frac{\tau_{by}}{\rho_{0}}\\ -\frac{1}{\rho_{0}}\left(\frac{\partial s_{yx}}{\partial x}+\frac{\partial s_{yy}}{\partial y}\right)+\frac{\partial }{\partial x}\left(hT_{xy}\right)+\frac{\partial }{\partial y}\left(hT_{yy}\right)+hv_{s}S \end{array}$$
where x, y are the Cartesian coordinates; u and v (m/s) are the components of velocity in the *x* and *y* directions; $\bar{u}$ and $\bar{v}$ (m/s) are, respectively, defined by $h\bar{u}=\int_{-d}^{\eta }udz$ $h\bar{v}=\int_{-d}^{\eta }vdz$; t (s) is the time; h (m) is the total water depth, h=η+d; η (m) is the height of the water surface relative to the undisturbed base surface, d (m) is the still water depth *; S* is the magnitude of the discharge due to point sources; *f* is the Coriolis parameter and is equal to 2ωsinφ (ω is the angular rate of revolution and φ is the geographic latitude); *g* is gravitational acceleration, *g* = 9.81 m/s^2^; ρ (kg/m^3^) is the density of water; $\rho_{0}$ (kg/m^3^) is the reference density of water); ($\tau_{sx}$, $\tau_{sy}$) (N/m^2^) and ($\tau_{bx}$, $\tau_{by}$) (N/m^2^) are the components of the surface wind and bottom stresses in the x and y directions; $p_{a}$ (Pa) is the atmospheric pressure; ($u_{s}$, $v_{s}$) is the velocity at which the water is discharged into ambient water.

#### 2.3.2. Habitat Suitability Index

Three indicators, the habitat suitability index (HSI), weighted usable area (WUA) and overall suitability index (OSI), were used to evaluate habitat quality. HSI is used to quantitatively describe the relationship between habitat preference and habitat factors [48]. The higher the habitat suitability value is, the higher the frequency of species. *OSI* [49,50] is the ratio of the *WUA* to the total computational domain area. For rivers with the best habitat quality, the theoretical *OSI* is 1, while 0 represents the worst habitat quality.
(4)$$HSI_{i}=SI(V_{i})\times SI\left(D_{i}\right)\times SI\left(S_{i}\right)$$
(5)$$WUA_{i}=\overset{n}{\underset{i=1}{\sum }}HSI_{i}\times A_{i}$$
(6)$$\mathrm{OSI}=\frac{1}{\sum_{i=1}^{n}A_{i}}\overset{n}{\underset{i=1}{\sum }}A_{i}HSI_{i}$$
where $SI(V_{i})$, $SI\left(D_{i}\right)$ and $SI\left(S_{i}\right)$ are the suitability of velocity, water depth and substrate, respectively. *HSI_i_* is the habitat suitability index of cell *i*. Each variable was quantified using a suitability curve between 0 and 1, with 0 indicating the lowest suitability and 1 indicating the highest; *A_i_* is the area of the *i*-th element, m^2^; *n* are the total number of elements in the study area.

In this study, by investigating the historical natural spawning habitats of *schizothoracids* in the Heishui River estuary and combining these data with those from other relevant studies of *S. prenanti* [51,52,53,54,55,56], the flow velocity and water depth preferences in different periods were determined (Figure 2). For *S. prenanti*, the suitable water depth range during the spawning period is 0.15~2.00 m, and the most suitable water depth is 0.20~0.80 m. The suitable flow velocity range is 0.07~2.00 m/s, the best flow velocity range is 0.20~0.8 m/s and the bottom material suitable for spawning is pebbles and gravel (diameter range is 2–256 mm). In the juvenile fish period, the suitable water depth range is 0.10~2.00 m, and the most suitable water depth is 0.40~1.20 m. The suitable flow velocity range is 0.00~1.00 m/s, the best flow velocity range is 0.10~0.50 m/s and the bottom material suitable for juvenile fish is pebbles and gravel (diameter range is 2–256 mm).

## 3. Results

### 3.1. Morphological Changes

Terrain monitoring was carried out in the study area before (2018), after (2019) and two years after (2021) dam removal. Limited by the monitoring conditions, only the river channel within the range of 300 m upstream to 400 m downstream of the dam site was measured in May 2019, and the terrain monitoring range in the other two periods was 1 km above the dam to 500 m below the dam.

The elevation change map of the river profile is drawn according to the measured topographic data of the study area, as shown in Figure 3. Topography monitoring results show that after dam removal, a large amount of sediment particles accumulated in the upstream reservoir are transported downstream, causing erosion of the upstream riverbed. Half a year after the dam was removed, the erosion depth in the reservoir area was approximately 0.5~2.93 m. Two years after the dam was removed, the riverbed erosion deepened, reaching 1.55~5.58 m. To minimize water pollution caused by sediment discharge, the riverbed near the dam site was manually dredged before dam removal, which is the main reason for the significant change in river morphology there. Half a year after dam removal, the riverbed at the dam site decreased by approximately 1.14 m, and two years later, it decreased by 3.67 m. Sediment deposition and erosion occurred in the riverbed downstream of the dam, and sediment deposition was more considerable. The sediment deposition was mainly concentrated in the river section 160 m~300 m downstream of the dam. The deposition height was approximately 0~1.17 m after half a year and 0~1.88 m after two years.

Figure 4 shows the elevation change in the riverbed two years after dam removal. The red area is the riverbed aggradation area. The darker the red is, the greater the aggradation. The blue area is the riverbed degradation area. The deeper the blue is, the greater the riverbed degradation degree is. Two years after dam removal, the riverbed upstream of the dam showed a general degradation trend, and the erosion of the main channel was clear. However, the left and right riverbeds immediately upstream of the dam site were raised by aggradation. These areas are vegetated floodplains where reduced flow velocity may be the main reason for sediment particle deposition. The sediment downstream of the dam was mainly deposited in the middle of the riverbed and left bank of the riverbed, and the riverbed on the right bank was clearly eroded. In the bend section 300 m~500 m downstream of the dam site, the river erosion depth on the left and right sides was relatively large, and the sediment deposition in the main channel was considerable. According to the field investigation, there is a relatively substantial central beach in this area, and the sediment deposition in the middle of the riverbed seems to be related to the central beach.

Figure 5 shows the morphological changes in typical sections before and after dam removal. The A–A’ section is located 600 m upstream of the dam site and is approximately 100 m wide (Figure 5a). After the dam was removed, there was marked sediment deposition on the right bank of the section, and the deposition height was between 0.6 and 1.5 m. The maximum erosion depth of the main channel was approximately 2.5 m. Section B–B’ corresponds to the dam site section and is approximately 60 m wide. After the dam was removed, the riverbed was degraded by approximately 1.4 m (Figure 5b). The riverbed erosion may have been influenced by the manual dredging, but, unfortunately, the specific dredging depth cannot be determined. The C–C’ section is located 500 m downstream of the dam site (Figure 5c). The channel on the right bank is obviously eroded, and the sediment is mainly deposited on the left bank, with a maximum deposition height of approximately 0.3 m. The specific positions of the three typical sections in the river are shown in Figure 4.

To study the influence of dam removal on riverbed materials, the change in average particle size before and after dam removal was compared and analyzed (Table 1). Limited by the scope of investigation, only two sections, 1 km upstream and 1 km downstream of the dam site, were investigated. The median particle size (D_50_) was measured at each cross-section to represent the sediment size of the reach. Sediments with particle sizes greater than 8 mm were not investigated before dam removal, and were determined by referring to the study of Tang et al. [6]. The median particle size of sediment after dam removal was determined according to the investigation results in spring 2021. The results show that the downstream sediment changed from coarse grain to fine grain, and the upstream sediment changed from fine grain to coarse grain. Before and after dam removal, the overburden layer of the upstream and downstream riverbed was mostly gravel or pebbles with diameters between 2 mm and 100 mm, which are suitable for the spawning and growth of *S. prenanti*.

### 3.2. Fish Habitat Quality Assessment

#### 3.2.1. Model Calibration and Validation

In the calculation of the two-dimensional fluid dynamics model, the upstream boundary is set as the flow provided by the automatic monitoring station of SXD, and the downstream boundary is the water level calculated according to the daily flow data of Ningnan hydrological station and the riverbed slope. The hydrodynamic models before and after dam removal were calibrated using the maximum water depth and maximum velocity data from the measured hydrological data of each survey section (Figure 1) in November 2018 (autumn) and March 2019 (spring), and the calculated roughness n in the model was 0.035~0.042 [57]. The comparison between model simulation results and actual measurement results is shown in Figure 6. The mean absolute error (MAE), root mean square error (RMSE) and R squared (R^2^) were introduced to the error analysis. The error analysis result is shown in Table 2, which indicates a good match between our model and the real river hydrodynamic conditions [58].

#### 3.2.2. Flow Characteristics

According to the simulation results, the influence of dam removal on the river flow pattern was compared and analyzed. The short-term changes in velocity and water depth after dam removal are shown in Figure 7. The river barrier point disappears, and the upstream overflow area decreases after the dam is removed. The water depth tends to decrease and varies from 0.04 to 1 m. The area near the dam site may be affected by artificial dredging, and the flow velocity increases significantly. The removal of the power station and diversion tunnel restored the natural flow of the river downstream of the dam and eliminated dehydration downstream. With the increase in discharge, the downstream water depth and velocity increased significantly, and the width of the water surface almost doubled, which increased the living space of aquatic organisms. According to the model calculation, the river area increased from 81,844.59 m^2^ to 84,745.06 m^2^, representing an increase of approximately 3.5%. After dam removal, the remolding of river morphology and the change in water flow distribution lead to the diversification of water flow structure, which promotes the formation of heterogeneous habitats, thus cultivating a rich river ecosystem [59].

#### 3.2.3. Physical Habitat

(1)Changes in habitat during the spawning period

Hydraulic variables such as flow velocity and water depth were correlated with the habitat requirements of *S. prenanti*. The habitat suitability distribution of *S. prenanti* during the spawning season is shown in Figure 8. After dam removal, the water depth suitability increases but the velocity suitability decreases. According to the analysis, the flow velocity is the main factor limiting the comprehensive suitability for *S. prenanti* spawning. As shown in Figure 8 and Table 3, after dam demolition, the WUA of *S. prenanti* in the upstream of the LMHD decreased from 47,146.39 m^2^ to 44,542.88 m^2^, a decrease of 5.52%.

The change in hydraulic conditions after dam removal has a significant impact on the downstream habitat. The recovery of the natural flow pattern increases the activity space of aquatic organisms and the suitable water depth area. After the dam was removed, the increased flow rate promoted the frequent exchange of matter and energy, making the water rich in oxygen. The high flow velocity and high dissolved oxygen content in the middle of the channel are not conducive to the adhesion of viscous fish eggs [60,61], leading to the gradual shift of suitable spawning habitats to both sides of the river bank. As seen from Table 3, the WUA of *S. prenanti* increased from 6206.9 m^2^ to 11,166.83 m^2^ in the downstream of the dam, with an increase of approximately 79.91%.

In terms of the overall suitable area of the river section during the spawning period, the WUA increased by 4.42% after the dam was removed. The removal of the dam improved the spawning habitat quality of *S. prenanti* in the study area, especially in the section downstream of the dam.

(2)Changes in habitat during the juvenile period

There is no apparent change in water depth suitability in the channel upstream of the dam after dam removal. However, the massive discharge of water and sediment reduces the area of the upstream river and the area suitable for juvenile fish. With the increase in flow velocity, the suitability of flow velocity decreases significantly, and flow velocity is the main factor limiting the comprehensive suitability. As shown in Table 3, the WUA of juvenile fish upstream decreased from 14,329.24 m^2^ to 11,984.44 m^2^, a decrease of 16.36%.

After dam removal, the natural flow was restored downstream of the dam. With the increase in water depth, the water depth suitability is significantly improved. As the flow velocity increased, the suitability of the flow velocity in the middle of the river channel decreased significantly, and the suitable growth area of juvenile fish gradually shifted from the river channel to the riparian slow-flow area. Figure 9 shows that after dam removal, the comprehensive suitability of juvenile fish growth downstream of the dam decreased significantly, and the flow velocity was the main indicator limiting their habitat suitability. The WUA of juvenile fish downstream of the dam decreased from 3599.93 m^2^ to 1155.49 m^2^, a decrease of 67.90%.

The changes in hydrodynamic conditions caused by dam removal had a great impact on the habitat quality of juvenile fish, and the overall WUA decreased by 26.71% in the study area.

#### 3.2.4. Habitat Sensitivity Analysis

Habitat sensitivities at the juvenile and spawning periods of *S. prenanti* were analyzed according to the WUA and OSI. It is noted that WUA and OSI had exactly the same trend, while OSI had different values at different critical periods of *S. prenanti* growth. Figure 10 shows that the OSI value of the spawning period is higher than that of the juvenile period. After dam removal, both the WUA and OSI in the spawning period increased from 29.78% to 31.09%, with an increase of 1.31%. The WUA and OSI decreased from 10.01% to 7.33%, with a decrease of 2.68% in the juvenile period. This shows that dam demolition has a positive impact on fish spawning but a negative impact on the growth of juvenile fish. From the above analysis, it can be seen that juvenile fish in this area are more sensitive to habitat changes.

## 4. Discussion

The change in hydrodynamic conditions caused by dam removal has different effects on the habitat quality of fishes in different growth periods. It can improve the spawning habitat but negatively affect the growth of juvenile fishes. The increase in flow velocity seems to be the main factor limiting habitat quality in this area. According to the calculation results, with the increase in the flow rate in the channel after dam removal, the suitable area for fish spawning is wider than the suitable growth area for juvenile fish, and the suitable habitat for juvenile fish is mainly distributed in the bank beach with slow flow velocity. This may be because juveniles do not have enough energy or muscle strength to develop in habitats with high flow velocity [62], and the low flow velocity on both banks can reduce energy losses during juvenile activity. In addition, juvenile fish need to accumulate a large amount of energy for growth and development, and the abundant bait resources in the river beach can provide the necessary energy for the growth of juvenile fish [63]. This suggests that shallow slow-flow areas may be an important survival area for juveniles [64]. In view of the characteristics of high flow velocity and shallow water depth in the southwestern mountainous areas of China, although the increase in velocity caused by dam removal can stimulate fish spawning, the number of fish stocks may not increase due to the low survival rate of juvenile fish under high velocity. In the early period of fish life, especially in the rapidly growing embryo and juvenile period, small changes in environmental factors may lead to developmental arrest of fish at this period, thus affecting the growth and development of individuals in the later period [65,66]. According to this analysis and previous work, the periods when fish are more sensitive to habitat changes, such as the juvenile and spawning periods, may be better suited for assessing environmental pressures on aquatic organisms.

Dam removal is a very important opportunity to restore the geomorphology and ecological function of river ecosystems. In the short term after dam removal, the river morphology upstream and downstream of the dam changes to different degrees. The headwater erosion of the upstream river makes the upstream river narrower and deeper, and the riverbed elevation decreases [67,68]. The large amount of fine sediment discharge makes the upstream sediment size coarser and the downstream sediment size finer [69]. The size changes downstream are minimal, probably due to fine sediment particles being moved further downstream by the current. Changes in river morphology and flow structure will further affect the distribution of deep pools and riffles in the river, diversifying the natural habitat structure and thus improving habitat quality in the river [6,7]. The river continuity restored by the removal of the dam allows migratory fish to normally use the habitat upstream of the dam site and to swim to suitable sites to spawn [70]. Dam removal increases fish migration and reproduction opportunities, but strategies to remove the LMHD may be limited in restoring river connectivity due to differences in the needs of different fish species. For fish with long-distance migration needs, the existence of the SXD will still restrict their migration and reproduction activities. In a follow-up study, whether the SXD and the two upstream dam sites need to be demolished to meet the needs of different protected fish needs to be further studied.

This study only considered the effects of velocity and water depth on the habitat suitability of *S. prenanti*. These two variables were chosen because they were assumed to be the most important variables to explain their habitat selection. Furthermore, these variables are used in physical habitat studies and are known as habitat templates [71,72]. In this study, the change in bed materials was analyzed based on field monitoring but using only the survey data of cross-sections. Since no model has been established to simulate bed material change, it is assumed that the change in the reach is similar to that in the section. *S. prenanti* prefer to spawn on pebbles and gravel at the bottom of rapids and move to shallow areas to feed after spawning, where juveniles generally live. While these qualitative descriptions are helpful, quantitative data on habitat substrates for *S. prenanti* are almost nonexistent. By comparing the changes in substrates, we believe that the changes in substrates are still within the suitable range of the target fish, so the effects of substrate changes on habitat suitability are not considered in this study. However, the substrate preference of fish in different growth periods is still a future research focus. Moreover, the environmental factors affected by dam removal are not limited to hydrodynamic parameters but also include the dissolved oxygen and water temperature [73,74]. According to the environmental factor requirements in different life periods, future studies should further consider other variables that may play an important role in habitat assessment to provide more comprehensive theoretical support for project management.

As an emerging river ecological restoration technology, the restoration effect of dam demolition is complex and uncertain. Habitat loss of fish at different growth periods may reduce the effectiveness of ecological restoration or even lead to ineffective restoration [3]. Therefore, successfully restoring the river ecological environment requires not simply dismantling dams but must also be combined with other river ecological restoration measures to carry out comprehensive river ecological management to achieve results [75]. Through the analysis, it can be seen that after dam removal, many parts of the downstream channel have become the main silting areas, and a large amount of sediment cover will affect the quality of fish spawning grounds until the river reaches a new balance. Therefore, in the section of the river with considerable silt accumulation, a T-shaped dam can be built after the flood season, and water diversion and sediment discharge can be carried out by improving the hydrodynamic conditions of the river section to achieve the purpose of further improving the quality the habitat of the fish. Planned stocking can promote the sustainable development of fish resources, restore the food chain and achieve long-term ecological restoration effects. The restoration and reconstruction of fish habitats can select sections of rivers with low flow rates as fish habitats and consider adding gravel, logs and aquatic weeds to the downstream riverbed to increase the growth environment of the fish. The protection of spawning grounds and nursery areas in rivers as part of subsequent fisheries management is also an effective way to increase the self-replenishment of fish stocks [76].

## 5. Conclusions

In China, many low-head dams are about to be removed due to poor management or economic efficiency. To explore the impact of low-head dam removal on the habitat suitability of *S. prenanti* in mountainous rivers, this study took LMHD removal in the Heishui River as an example and combined field survey data with numerical simulation results to study the changes in river morphology and fish habitat after dam removal, and drew the following conclusions.

After dam removal, the change in water and sediment conditions leads to the erosion and deposition of particles upstream and downstream of the dam, respectively. The riverbed material upstream becomes coarse, and the riverbed material downstream becomes fine. However, the riverbed material before and after dam removal is within the size range of coarse-grained stone. The change in river morphology affects the hydraulic conditions in the river to a certain extent, which may change the existing distribution of deep pools and shoals, increase the diversity of riverbed geomorphology, and further affect the quality of the biological habitat in the river.

The habitat suitability calculated by the model shows that with the restoration of natural hydraulic conditions, the quality of the spawning habitat is significantly improved in the downstream reaches and decreased in the upstream reaches. For the juvenile period, the habitat quality of the upstream and downstream reaches of the dam site tends to decrease, which is not conducive to the growth and development of juvenile fish. The dam removal restored the continuity of the river habitat, and the WUA increased by 4.42% during the spawning period and decreased by 26.71% during the juvenile period throughout the reach. The influence of hydraulic conditions on the juvenile period was greater than that on the spawning period, and the flow velocity was the main limiting factor for the quality of the fish habitat in this area.

In general, our research results provide a new idea for dam removal management. According to the needs of fish in different growing stages, the method of dam removal can be discussed and designed in detail, which can minimize the disturbance of the fish population caused by the change in hydrodynamic conditions in the river. However, it is also crucial to track and monitor the changes in fish stocks after dam removal because the changes in biological resources can more intuitively reflect the link between fish reproduction and the benefits of barrier removal. What is more, the subsequent protection or restoration of the habitat should also be emphasized to ensure that dam removal can achieve the best effect and provide theoretical support for the adaptive management of restoration measures.

## Figures and Tables

**Figure 1 ijerph-19-11743-f001:**
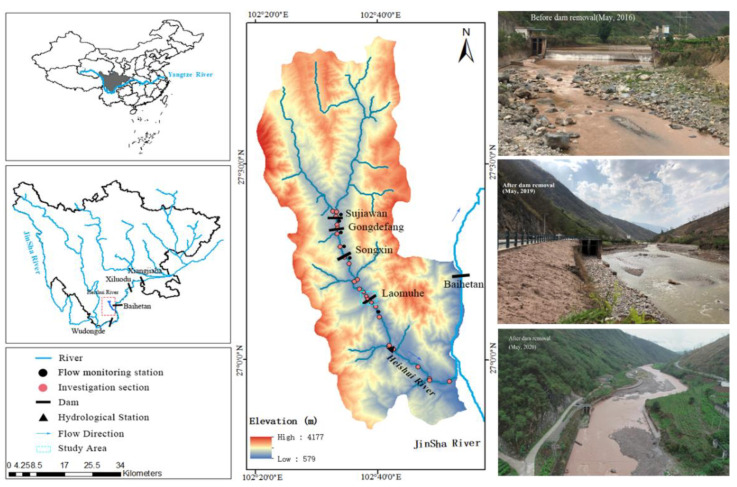
Location (**left**) and basin elevation (**center**) of the Heishui River. The picture shows the river reach near the dam site before (2018, **top right**), during (2019, **middle right**) and two years after (2021, **bottom right**) removal.

**Figure 2 ijerph-19-11743-f002:**
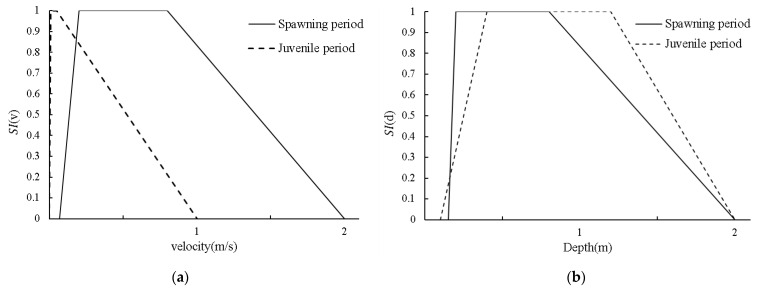
Suitability index (*SI*), (**a**) suitability index of velocity (*SI(v**)*), (**b**) suitability index of depth (*SI(**d)*).

**Figure 3 ijerph-19-11743-f003:**
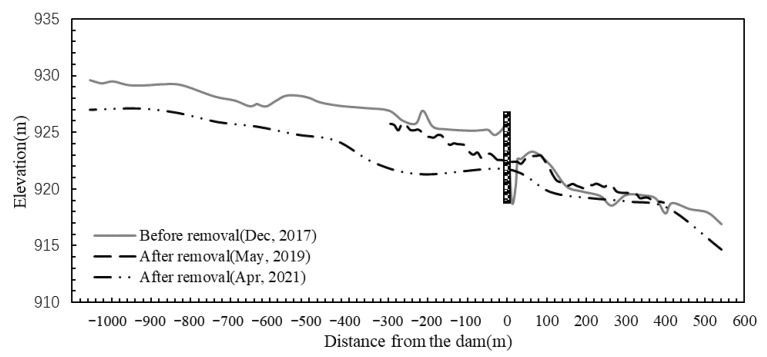
Changes in the longitudinal profiles of elevation with dam removal.

**Figure 4 ijerph-19-11743-f004:**
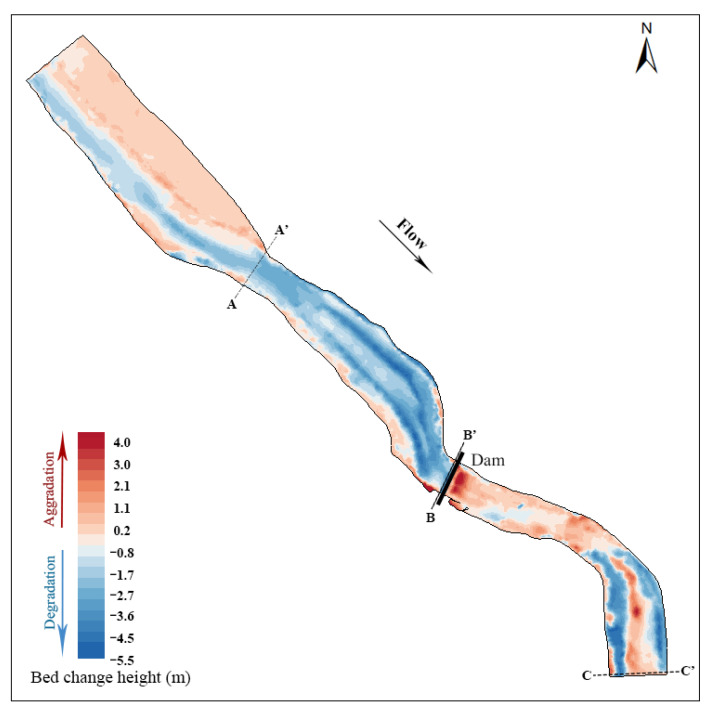
Bed elevation change before and after dam removal.

**Figure 5 ijerph-19-11743-f005:**
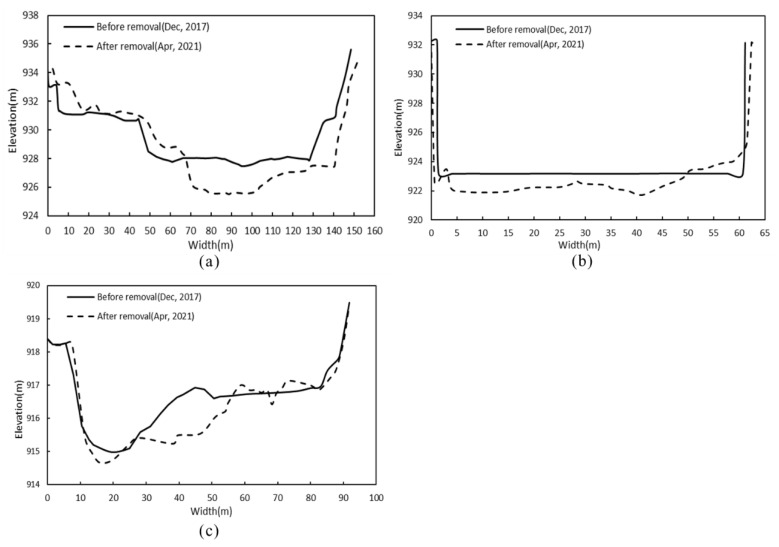
Changes in the channel cross-section after dam removal. The positions of (**a**) section A–A’, (**b**) section B–B’ and (**c**) section C–C’ are shown in Figure 4. The water flow direction is vertical and faces outward.

**Figure 6 ijerph-19-11743-f006:**
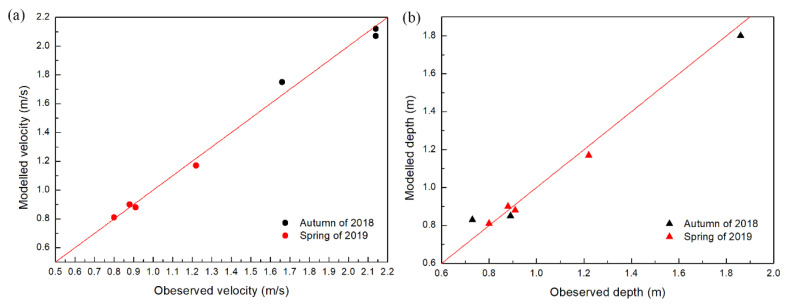
Comparison of observed velocity and water depth data with modelled results: (**a**) flow velocity and (**b**) water depth. The magnitude of the values varies with the flow in different periods.

**Figure 7 ijerph-19-11743-f007:**
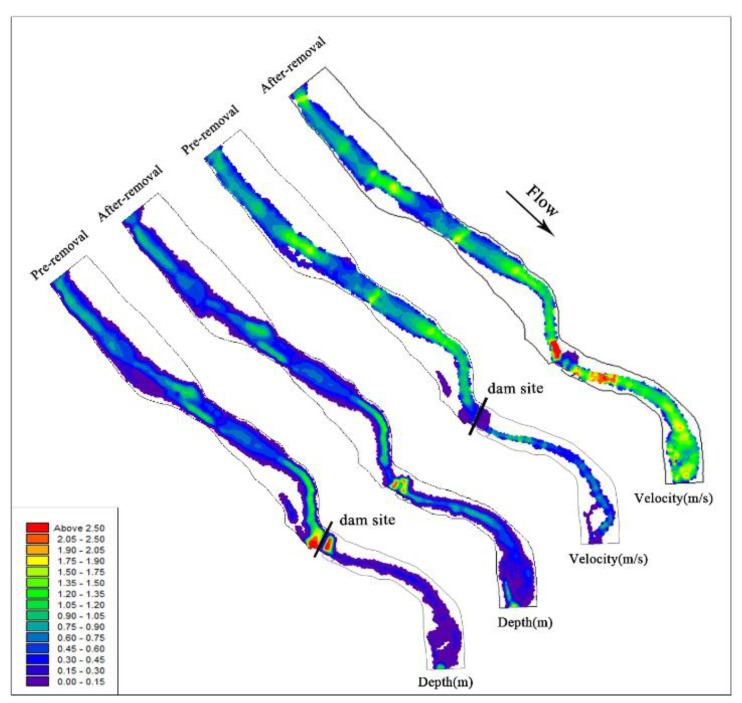
Velocity and water depth distribution map of the river before and after dam removal.

**Figure 8 ijerph-19-11743-f008:**
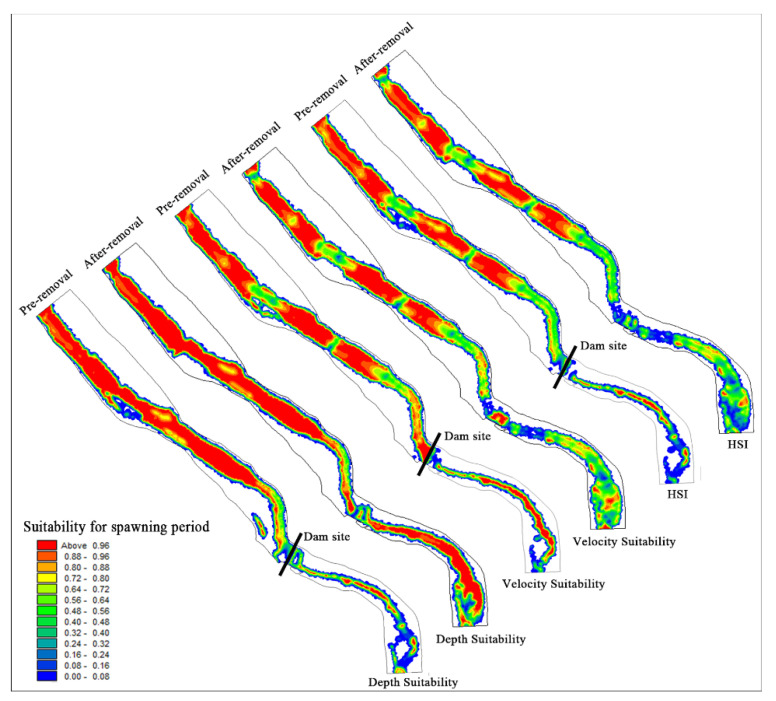
Distribution map of velocity suitability, water depth suitability and habitat suitability before and after dam demolition during the spawning period.

**Figure 9 ijerph-19-11743-f009:**
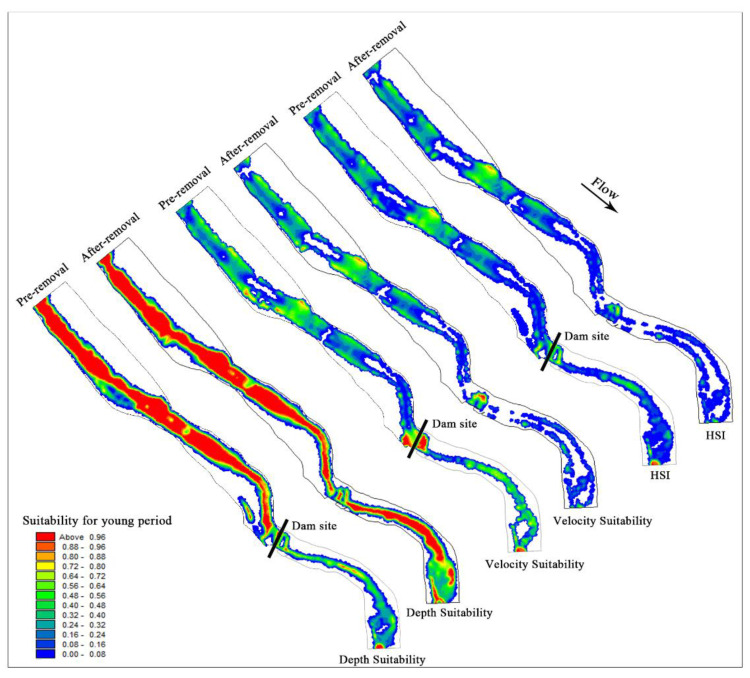
Comparative analysis of velocity suitability, water depth suitability and comprehensive suitability before and after dam demolition at the juvenile stage.

**Figure 10 ijerph-19-11743-f010:**
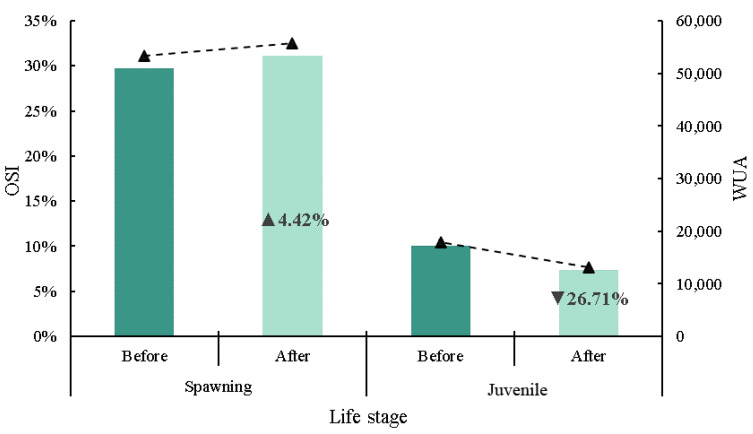
Changes in OSI (overall suitability index) and WUA (weighted usable area) values for *S. prenanti* before and after dam removal. ▲: increase and ▼: decrease.

**Table 1 ijerph-19-11743-t001:** Variation in median substrate size (D_50_).

Median Grain Size (mm)	Before Dam Removal [6]	After Dam Removal
Upstream of the Laomuhe Dam	37.05	87.14
Downstream of the Laomuhe Dam	75.83	73.33

**Table 2 ijerph-19-11743-t002:** Error analysis of velocity (V) and depth (D) simulation results.

Season		RMSE		R^2^		MAE	
		V (m/s)	D (m)	V (m/s)	D (m)	V (m/s)	D (m)
Pre-removal	Autumn of 2018	0.07	0.07	0.98	0.99	0.060	0.067
After removal	Spring of 2019	0.08	0.03	0.92	0.99	0.065	0.028

**Table 3 ijerph-19-11743-t003:** Changes in weighted usable area (WUA) values for *S. prenanti* before and after dam removal.

Flow Rate (m^3^/s)	Life Stage	Condition	Weighted Usable Area (WUA/m^2^)		
			Upstream Reach	Downstream Reach	Total Reach
23.55	Spawning	Before	47,146.39	6206.90	53,353.29
		After	44,542.88	11,166.83	55,709.71
		Change Rate	▼5.52%	▲79.91%	▲4.42%
	Juvenile	Before	14,329.24	3599.93	17,929.17
		After	11,984.44	1155.49	13,139.92
		Change Rate	▼16.36%	▼67.90%	▼26.71%

Note: ▲: increase and ▼: decrease.

## Data Availability

Not applicable.

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
