# Peer review of "Impact of Low-Head Dam Removal on River Morphology and Habitat Suitability in Mountainous Rivers"

_ijerph, 2022, doi:10.3390/ijerph191811743_

Round 1

Reviewer 1 Report

The paper aimed to evaluate the Heishui River, in China, before and after remove the dam, considering the impacts into two growth periods: the spawning and juvenile phases of S. prenanti.

Several metrics were made considering the Flow characteristics, WUA (Weighted Used Area), depth and velocity. Investigations were conducted however, I did not understand with details the methodologic schemes.  

I recommended the paper for publication, because the scope of this study is really interesting, however, it is need to improve the writting and get a better understanding, mainly in Sections 3.2.2 and 3.2.3

Besides, the authors need to correct some observerd details:

p.1-L40: IUCN – define the acronym

p.2-L64 – quality of habitat?

Figure 1 – Needs to be translated into English, mainly the Elevation map

p.6 L204 – The errors are showed in Table 1 (not in Figure 3, as metioned).

p.8 L260 - Error in Reference of Figure 6.

Author Response

We would like to thank the editor and reviewers for their helpful comments on our manuscript.  We have studied their comments carefully and made a comprehensive revision on our manuscript (in Red). We have attached the point-to-point reply. Please see the attachment for details.

Reviewer 2 Report

The study presented in article is a continuation and extension of the authors research focusing on the impact of external environmental changes having Laomuhe Dam removal project of the Heishui River as example. The authors studied the impact of dam removal on river morphology and fish critical habitat within short term, by incorporating juvenile and spawning habitat requirements into the ecological impact assessment of dam removal by coupling the numerical simulation and field survey methods.

The study provides valid data. 

Specific comments:

Introduction: It is comprehensive and generally correct.

L104: ”three years”? In M&M L140 you mentioned ”four years”.

Materials and methods: Must be improved

L122: For a better localization, GPS coordinates of LMHD could be useful to be added

L144: What materials and methods did you used for investigation of the „ Fish, plankton, epiphytic algae and benthic animal resources”. Did you use electrofishing? … plankton net … etc. A short description could be useful.

L142: How the 17 monitoring sections were chosen? Equal distances among them, or according to the stream shape …. How long is the total distance between first and last monitoring sections?

L146: It is not clear where the 2 biological resource monitoring sites (SC1 and SC2) are located. Are they in some of the 17 monitoring sections? Please clarify.

L148-158: The sentence has to be moved to Discussion. This do not contain M&M

L174: reference?

L182-192: The sentence has to be moved to Discussion. This do not contain M&M

L203-210: These do not contain M&M. Could be better to be moved at results

Results, discussions and conclusions: are generally correct

Author Response

(The authors gave the same response as above.)

Reviewer 3 Report

Dear authors,

The title manuscript: Impact of low-head dam removal on river morphology and habitat suitability in mountainous rivers. The study coupled field survey data and two-dimensional hydrodynamic model to explore the changes in river morphology at different scales and the impact of changes in hydrodynamic conditions on fish habitat suitability in the short term. The manuscript uses hydraulic data from other studies (References [45, 48]) to perform three-dimensional simulations. Although the authors describe that biological data collections were performed (Chapter 2.2) they were not correlated with the results of the manuscript. In this way, this description could be withdrawn.
The authors should highlight in the manuscript that the hydraulic data and the results of the study refer only to the fish species - Schizothorax prenanti.
My suggestions of corrections and questions are in text boxes, in the file attached to the review.

Author Response

(The authors gave the same response as above.)

Round 2

Reviewer 1 Report

Modifications were made. That's work for me.

Congratulations.

Author Response

Thanks to the editors and reviewers for their valuable comments. We are very grateful to hear the reviewers' recognition and affirmation of our work. We sincerely wish you success in your work!

Reviewer 2 Report

The revised manuscript has been adequately improved. The authors incorporated the main suggested information. The critical points were reached. The previous version have been correctly modified.

Author Response

(The authors gave the same response as above.)

Reviewer 3 Report

Dear authors,

You performed in an exemplary manner all the corrections and adjustments recommended by me. I've only included three small corrections (see legends of Figures 2, 6 and 10, in the text box - attached file), just to finalize my review.

Author Response

We would like to thank the reviewers for their recognition of our work and the editors and reviewers for their valuable comments on our manuscript. We have carefully studied their reviews and have fully revised our manuscript (in red). We have attached the point-to-point replies. See the attachment for details.
